# Numerical and Experimental Study of the Mechanical Response of Diatom Frustules

**DOI:** 10.3390/nano10050959

**Published:** 2020-05-18

**Authors:** Emre Topal, Hariskaran Rajendran, Izabela Zgłobicka, Jürgen Gluch, Zhongquan Liao, André Clausner, Krzysztof Jan Kurzydłowski, Ehrenfried Zschech

**Affiliations:** 1Dresden Center for Nanoanalysis, Technische Universität Dresden, 01069 Dresden, Germany; ehrenfried.zschech@ikts.fraunhofer.de; 2Fraunhofer IKTS, Institute for Ceramic Technologies and Systems, 01109 Dresden, Germany; rharishankaran@gmail.com (H.R.); juergen.gluch@ikts.fraunhofer.de (J.G.); zhongquan.liao@ikts.fraunhofer.de (Z.L.); andre.clausner@ikts.fraunhofer.de (A.C.); 3Faculty of Mechanical Engineering, Bialystok University of Technology, 15-351 Bialystok, Poland; i.zglobicka@pb.edu.pl (I.Z.); k.kurzydlowski@pb.edu.pl (K.J.K.)

**Keywords:** X-ray computed tomography, biomaterial, 3D morphology, diatom, finite element analysis

## Abstract

Diatom frustules, with their hierarchical three-dimensional patterned silica structures at nano to micrometer dimensions, can be a paragon for the design of lightweight structural materials. However, the mechanical properties of frustules, especially the species with pennate symmetry, have not been studied systematically. A novel approach combining in situ micro-indentation and high-resolution X-ray computed tomography (XCT)-based finite element analysis (FEA) at the identical sample is developed and applied to *Didymosphenia geminata* frustule. Furthermore, scanning electron microscopy and transmission electron microscopy investigations are conducted to obtain detailed information regarding the resolvable structures and the composition. During the in situ micro-indentation studies of *Didymosphenia geminata* frustule, a mainly elastic deformation behavior with displacement discontinuities/non-linearities is observed. To extract material properties from obtained load-displacement curves in the elastic region, elastic finite element method (FEM) simulations are conducted. Young’s modulus is determined as 31.8 GPa. The method described in this paper allows understanding of the mechanical behavior of very complex structures.

## 1. Introduction

Biomimetics and bioinspired materials are emerging fields for learning design strategies from nature and applying them to synthetic structures. Through millions of years of evolution, nature emerged into unique designs that even surpass modern engineering solutions [1,2,3,4,5,6]. A deep understanding of such natural materials would provide invaluable insight for the design of advanced synthetic materials. Diatoms are unicellular, eukaryotic, photosynthetic microalgae which ubiquitously appear in almost every aquatic environment on Earth [7]. Diatoms are one of the surprisingly elegant example of nature’s sophisticated design skills, demonstrating how organisms build refined structures based on a bottom-up approach and simple components. Their hard porous shell, also known as frustule, is composed mainly of amorphous bio silica which possesses remarkably organized and hierarchical 3D porous exoskeleton structures at the nano, submicrometer, and micrometer scales [8,9,10,11]. These hierarchical porous structures give rise to the frustules’ extraordinary mechanical properties [12,13], very large surface area [14], and unique optical properties [15,16,17].

The common explanation is that the frustules have largely evolved as responses to specific predator types and functions as a protective armor for the cells [12]. Thus, the nature’s motivation for using an hierarchical arrangement of several structural elements at carefully chosen varied length scales may have evolved out of the obvious advantage of optimizing the mechanical properties [18,19]. Mechanical properties of cellular materials are usually defined by the unit-cell geometry, the relative density and the solid material properties [20]. The general assumption in the classical theory of fracture mechanics of cellular solids is that the properties of the parent solid are independent of its dimensions. That means, cellular solids with the identical skeleton material and pore topology should have the same Young’s modulus and strength, regardless of their absolute dimensions [19,20]. However, this assumption may not reflect the mechanical properties of hierarchical porous biological structures which have been described through property improvement beyond the rule of mixtures [19,21,22]. The introduction of size effects in the mechanical strength of nano-scale solids (<100 nm) might play a critical role for the mechanical properties of hierarchical porous biological structures [23]. Therefore, in order to understand and predict the structural response of structures that contain micro- and nano-scale components, as in the case of frustule and other hard biological materials, size-dependent mechanical properties of constituent materials need to be considered and to be incorporated into models. In particular, considering frustules, molecular dynamics simulations of a simple model of generic nano-porous silica structures have shown that below a critical length scale around 6–8 nm, the silica struts in nano-porous structures undergo plastic shear deformation before fracture, leading to enhanced ductility [24]. The size effect investigations in silicon nanowires through atomistic simulations shown an increased stiffness if the diameter of silicon nanowires is smaller than 6 nm [25,26].

The mechanical behavior of frustules was studied using micro- and nanomechanical testing as well as finite element analysis (FEA) [27,28,29,30,31]. Atomic force microscopy (AFM) nanoindentation measurements of frustules revealed that Young’s modulus and hardness values are varying significantly, from 7 to hundreds of GPa, and from 1 to 12 GPa, respectively, depending on the location where the measurement was performed [27,32]. These findings support that the mechanical performance is not only related to the amorphous bio silica itself but also to the hierarchical 3D morphology of porous exoskeleton structures at the nano, submicrometer to micrometer scales of the frustule. FEA is a favored approach to study mechanical properties of biomaterials, especially to understand complex hierarchical 3D porous structures [12,19,22,28,29]. In the FEM framework, the general approach is to simplify the geometry to reduce computational efforts and to increase the likelihood of convergence of the numerical model. However, the drawback of this straightforward approach is that the final model does not reflect the real morphology, and thus it is not applicable for modelling a local stress distribution. However, image-based FEA approaches such as our novel extremely detailed X-ray computed tomography (XCT)-based FEA, offer a unique way to replicate the real morphology with preserving even the finest features. Obvious disadvantages of this approach are the numerical efforts and the requirement of high-performance computing to solve this complex problem. In the literature, the image-based FEM is extensively used to study mechanical properties of bone microstructures [33,34]. However, these studies were at limited scale and not combined with precise mechanical investigations conducted at the same sample.

Scanning electron microscopy (SEM) and AFM were also extensively used to reveal morphologies and structures of the diatom frustule [8,11,27,32,35]. However, these techniques are limited to outer surfaces, fracture surfaces, and cross-sections, restricting the observations of the interior structures in 3D. Thus, to correlate the morphological effects accurately to the mechanical properties, a truly 3D characterization method is required. The focused ion beam (FIB) serial cutting technique and subsequent imaging of a series of cross-sections in the SEM provides high-resolution 3D morphological information. It was successfully demonstrated that the FIB-SEM technique is a powerful tool to investigate the internal structures of frustule [10,36,37]. However, the main drawback of this technique is its destructive nature. In addition, ion beam milling of the hierarchical materials can cause artefacts. The nano-XCT technique provides the 3D morphology of hierarchically structured materials nondestructively with sub-100 nm resolution [10]. Furthermore, the obtained 3D information was transferred into a computer-aided design (CAD) file and it was 3D printed as an up-scaled, artificial, self-similar object. A proof-of-concept of the workflow to engineer objects that are self-similar to natural objects was demonstrated for a diatom frustule structure [38].

To establish the 3D morphology-mechanical properties relationship for the *Didymosphenia geminata* frustule, which has a pennate symmetry, a systematic approach was developed and applied that employs 3D visualization, in situ mechanical characterization, and FEA simulation for the same sample. The experimentally obtained load-displacement data are used as input for FEM simulations. Furthermore, the real 3D morphology data of the frustule obtained by nano-XCT provide the basis for generating a frustule model. This model is used for the simulation of the nano-scale indentation into the material and enabled an access to the load-displacement-data of the simulated experiments after an iterative error minimization cycle. This work reports first time a generated “digital twin” model of the experimentally studied sample for numerical simulations. The method described here holds a great potential to understand the mechanical behavior of complex structures such as in biomaterials, catalysts, and other porous or skeleton materials.

## 2. Materials and Experimental Setup

### 2.1. Structure Characterization

A laboratory nano-XCT tool (Xradia nano-XCT-100, Xradia Inc., Pleasanton, CA, USA) in large field of view in phase contrast mode was used at a photon energy of 8 keV. An intact frustule was fixed on the tip of a pin stub mount that is compatible with SEM and in situ nanoindenter adapters too. The field of view was 66.5 × 66.5 µm^2^ with 512 × 512 pixels, resulting in a voxel size of 0.13 µm. Two scans were done to cover the whole object and to obtain 3D information of a single complete *Didymosphenia geminata* frustule. The tilt series for the tomography consisted of 401 images within an angular range of 180°. The exposure time per image was 120 s. The acquired projections were reconstructed using a Filtered Back Projection (FBP) algorithm [39]. The applied reconstruction methodology provides highly accurate 3D morphological data by enabling corrections for imaging artefacts of high-resolution XCT though the compensation of acquisition inaccuracies such as misalignment and motions of samples and tool components [40,41]. The reconstructed image stacks are fused into one stack that includes whole 3D morphological information of the frustule.

Because of the spatial resolution that was achieved in this nano-XCT study, structures smaller than 130 nm could not be captured. Therefore, a SEM study was conducted using a Dual Beam SEM-FIB system (FEI Helios NanoLab 660, Thermo Fisher Scientific, Waltham, MA, USA), operating at an accelerating voltage for the electrons of 1.0 kV. The transmission electron microscopy (TEM) lamella samples were prepared using a Dual Beam SEM-FIB system (Carl Zeiss NVision 40, Carl Zeiss AG, Oberkochen, Germany) and a TEM study is conducted using a scanning TEM (Carl Zeiss Libra 200 MC Cs, Carl Zeiss AG, Oberkochen, Germany), operating at an accelerating voltage of 200 kV, were conducted to image nano-structure, nano-porosity, and elemental distribution in raphe and rib locations of another *Didymosphenia geminata* frustule sample.

### 2.2. Mechanical Characterization

A microindenter system (Hysitron/Bruker Picoindenter PI87, Bruker/Hysitron, Minneapolis, MN, USA), as a micromanipulator in a Dual Beam SEM-FIB system (FEI Helios NanoLab 660, Thermo Fisher Scientific, Waltham, MA, USA), was used to characterize the micromechanical behavior of the *Didymosphenia geminata* frustule after imaging its 3D morphology using nano-XCT. The application of the microindenter system in a SEM-FIB system allows the performance of in situ indentation experiments on specific locations of the frustule in order to obtain local material response information. A conical indenter with a spherical tip with 3.2 µm radius was used. The applied load was varied between 100 µN and 1400 µN. The SEM images were recorded during the indentation experiments, and the load versus displacement data were extracted.

### 2.3. Finite Element Analysis

The 3D reconstructed XCT data were used as input for creating a “digital twin” of the mechanically investigated *Didymosphenia geminata* frustule. The reconstructed 3D image stack was segmented and converted into a surface mesh using the Simpleware ScanIp software (Synopsys, Inc., Mountain View, CA, USA) [42], which enables the preservation of all features of the frustule. Subsequently, a reverse engineering approach using the SpaceClaim tool (ANSYS, Inc., Canonsburg, PA, USA) [43] was applied to synthesize the geometry of the frustule. This approach provides a high flexibility for building several models that require different simulation setups. A tetrahedral mesh was generated using the FEM software ANSYS (ANSYS, Inc., Canonsburg, PA, USA) [44], and a nonlinear geometry solver was implemented to capture large deflections of the structure. The final mesh consisted of 21.6 million tetrahedral elements and 1.2 million hexahedral elements with an average mesh quality of 0.95. The load-displacement data obtained through in situ micro-indentation were used to obtain accurate material data for the FEM simulations using an isotropic elastic material model. The micro-indentation experiments in the SEM are replicated in the FEM framework to describe the mechanical properties at several structure levels of the frustule. The applied methodological workflow is provided in Figure 1.

## 3. Results and Discussion

### 3.1. Structural Characterization

The virtual cross-sections from reconstructed nano-XCT data of the *Didymosphenia geminata* frustule are shown in Figure 2. Nano-XCT data, and the derived virtual cross-sections, resolve the frustule including raphes, ribs, and stigmas. The frustule morphology shows a bilateral symmetry for top (epivalve) and bottom (hypovalve) valves. The epivalve part is considerably larger than the hypovalve part. The difference in the width of the frustule ends (apices) suggests that there is no bilateral symmetry over the valve.

To evaluate smaller interior structures, SEM and TEM investigations were conducted on another *Didymosphenia geminata* sample. The SEM used has an image resolution of 0.7 nm at 1 kV electron beam acceleration voltage. The SEM images of inner ribs and center raphe parts exhibit a high level of structural openings (Figure 3). The images of the fine strip structures on the apex region—the end of frustule (160–210 nm width)—confirm the results achieved using nano-XCT.

Since chemical composition and porosity determine the mechanical properties of porous structures, the inner rib and the center raphe structures were studied using TEM, and an energy-dispersive X-ray spectroscopy (EDX) analysis was performed to generate the elemental maps of these structures. The TEM images in Figure 4 show lots of nano pores, especially in the central raphe region. The average pore diameter was determined as 14.8 nm.

EDX maps of internal ribs and central raphe regions reveal the elemental composition of the inner structures of the frustule (see Figure 5). The data show that silica is the major component of the frustule. No gradients in the elemental distribution along the inner skeleton parts could be detected.

The study at this particular frustule demonstrates that nano-XCT provides 3D morphological and structural information from a complete frustule nondestructively, which is unique compared to other analytical techniques. The resolved structural information was confirmed based on SEM and TEM data. The 3D morphological and structural information obtained by the multi-scale microscopy approach (X-ray and electron microscopy techniques with different field of view and spatial resolution) is adequate to construct a full FEA model of the cell walls with periodic lattice structures from a *Didymosphenia geminata* frustule.

### 3.2. Mechanical Characterization

The mechanical characterization was conducted on the *Didymosphenia geminata* frustule sample subsequently to the nano-XCT investigation. To obtain the local material response using a spherical indenter tip, micro-indentation experiments were performed at seven different locations, following the spot numbering order, on the epivalve surface. The applied load range was between 100 µN and 1400 µN. The obtained load-displacements curves for each micro-indentation spot are given in Figure 6. During the micro-indentation experiments, SEM images were recorded continuously to track the deformation. An elastic deformation along with residual displacement caused by the damage on interlocking structures of frustule was observed. The obtained load-displacement curves confirmed that the residual displacement was less than 200 nm under a load of 600 µN. For higher loads, the behavior was similar. However, the residual displacement was in the range of 300 nm and 1.5 µm, with an obvious creep behavior. The observed non-linearities in the load-displacements curves can be explained as a result of deformation in the interlocking structures.

In case of only elastic response of the material during indentation, the elastic stress field at the contact region is well defined regardless of the type of indenter. However, for the elastic response as observed for the *Didymosphenia geminata* frustule, to obtain stress fields around the contact area is an extremely complex task. Especially considering the 3D complex structures of the *Didymosphenia geminata* frustule, an analytical solution cannot be obtained easily. Therefore, the mechanical behavior is studied in this paper by applying elastic FEM simulations in addition.

### 3.3. Finite Element Analysis

Since the *Didymosphenia geminata* frustule has complex structural features, the use of a simplified geometry for the FEM simulations will not provide accurate simulation results. Moreover, due to its structural complexity it is not an easy task to design a representative complete *Didymosphenia geminata* frustule geometry in a computer-aided design (CAD) framework even if the exact feature sizes are known. Therefore, 3D data obtained from the nano-XCT study were used to construct a geometry based on the real *Didymosphenia geminata* frustule sample. The segmented label image and the 3D visualization of the reconstructed nano-XCT data together with generated surface mesh is provided in Figure 7. The segmentation was performed using a simple threshold method. Finally, the obtained label fields on each slice were cleared manually to ensure that there are no errors introduced due to mis-segmentation. The obtained surface mesh enabled preservation of all details and the construction of a CAD geometry that was modifiable for different simulation cases, which is one of the main advantages of our approach.

The obtained FEA model and the boundary conditions are given in Figure 8. The use of a real sample allowed us to replicate the micro-indentation experiments. The experimentally obtained load versus displacement data were used as input for the FEM framework to estimate the local stresses within the structure for the applied load. A purely elastic material model is used to reflect the negligible plasticity and deformation due to limited residual displacements observed in the micro-indentation experiment. In the FEM simulation, the central raphe of *Didymosphenia geminata* frustule was considered, i.e., spot 7. This region was selected because the effect caused by deformation in the interlocking structures on load versus displacement curves was minimal.

In the FEM framework, the Young’s modulus was used as a free parameter to minimize the deviation between the simulation and the indentation experiment and subsequently, inversely determine the real Young’s modulus of the frustule. For the simulations, the Poisson’s ratio is selected as 0.17 [32]. The initial Young’s modulus was 22.4 GPa [13]. An iterative optimization function was used to predict the Young’s modulus as a function of maximum displacement. 12 iterations were performed for the load of 1200 µN. The deviation of the Young’s modulus and the maximum von Misses stress at 2.5% equivalent von Misses strain are shown in Figure 9. For the stress determination, 2.5% strain is selected since the failure strain for frustules are reported between 2.2% and 4.0% [13].

The comparison of the obtained load-displacement curve at the end of iteration cycle with the micro-indentation experiment under the load of 1200 µN is given in Figure 10. Besides the non-linearities observed in the load-displacement curve from the experiments, both curves were consistent. The applied iterative parameter deviation approach in the FEM framework enabled the determination of the Young’s modulus of the frustule as 31.8 GPa for almost the same maximum displacement as in the indentation experiment. Compared to amorphous bio silica that has a Young’s modulus of about 70 GPa, the frustule has a significantly lower Young’s modulus which can be explained by the porosity. The frustule shows elastic deformation even for high deformation that allows the regaining of its original shape.

## 4. Conclusions

In this study, a methodological approach that combines structural, mechanical, and numerical characterization was applied to extract micromechanical properties of a *Didymosphenia geminata* frustule. An intact frustule was studied using nano-XCT to obtain 3D morphological and structural information which subsequently was used to generate a “digital twin” model of the experimentally studied sample for numerical simulations. Nano-XCT investigations enabled resolution of the inner structures of *Didymosphenia geminata* frustule nondestructively with a resolution of 130 nm. SEM and TEM studies conformed the hierarchical structures of the *Didymosphenia geminata* frustule determined with nano-XCT. That means, nano-XCT is a powerful tool to obtain 3D information from a frustule nondestructively. However, electron microscopy was needed to image nano-scale pores. In situ micro-indentation experiments were conducted to measure the local mechanical response at several locations. An elastic response under the spherical indentation was concluded from the experimentally obtained load-displacement curves. Based on the newly developed and applied methodology, mechanical properties of the *Didymosphenia geminata* frustule were determined. The Young’s modulus of investigated *Didymosphenia geminata* frustule was estimated as 31.8 GPa. The combined theoretical and experimental approach demonstrated here holds a great potential to extract the mechanical properties of complex structures, particularly of porous and skeleton materials. The future work will focus on higher resolution 3D data to allow the generation of a unit-cell model of the *Didymosphenia geminata* frustule, with the goal to isolate the contribution of each component.

## Figures and Tables

**Figure 1 nanomaterials-10-00959-f001:**
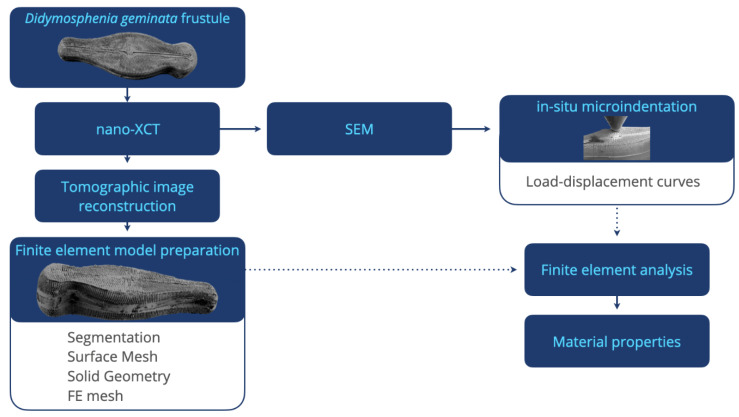
The methodological workflow used to extract material properties of *Didymosphenia geminata* frustule.

**Figure 2 nanomaterials-10-00959-f002:**
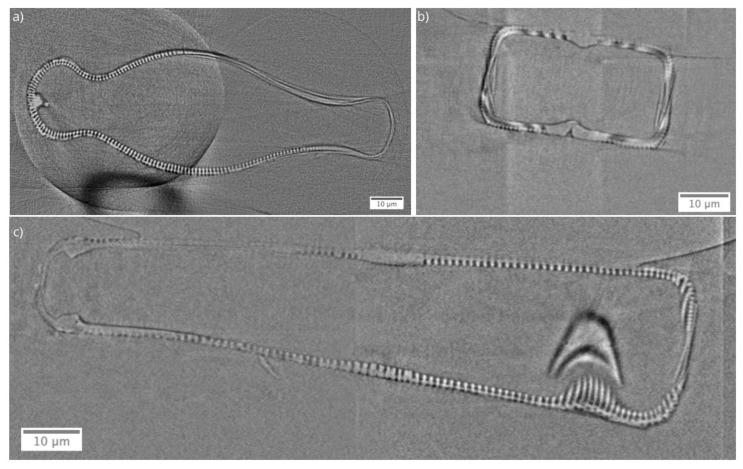
Virtual cross-sections from reconstructed nano-X-ray computed tomography data of the *Didymosphenia geminata* frustule. (**a**) XY slice, (**b**) XZ slice, and (**c**) YZ slice.

**Figure 3 nanomaterials-10-00959-f003:**
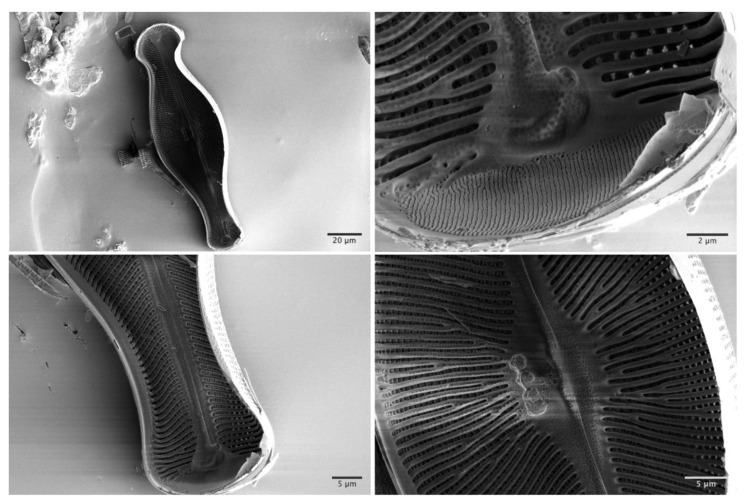
Scanning electron microscopy images of inner structures of *Didymosphenia geminata* after the epivalve part was separated from the hypovalve part.

**Figure 4 nanomaterials-10-00959-f004:**
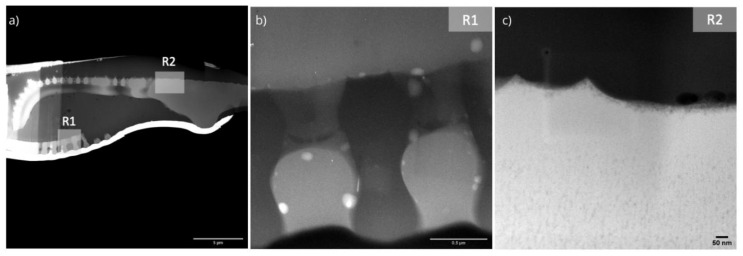
Scanning transmission electron microscopy images of a *Didymosphenia geminata* frustule sample which prepared by a FIB procedure: (**a**) overview of the lamella (dark field image, scale bar 5µm), (**b**) internal ribs (bright field image, scale bar 0.5 µm), and (**c**) central raphe (dark field image, scale bar 50 nm).

**Figure 5 nanomaterials-10-00959-f005:**
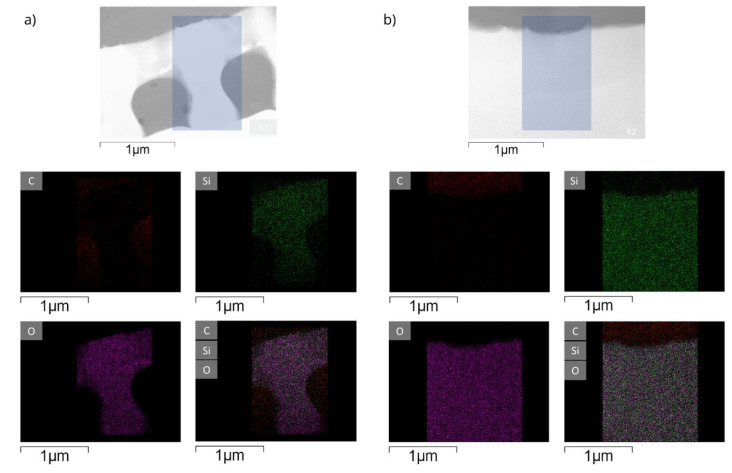
Energy-dispersive X-ray spectroscopy maps for carbon, oxygen and silicon of (**a**) internal ribs, and (**b**) central raphe of a *Didymosphenia geminata* frustule sample.

**Figure 6 nanomaterials-10-00959-f006:**
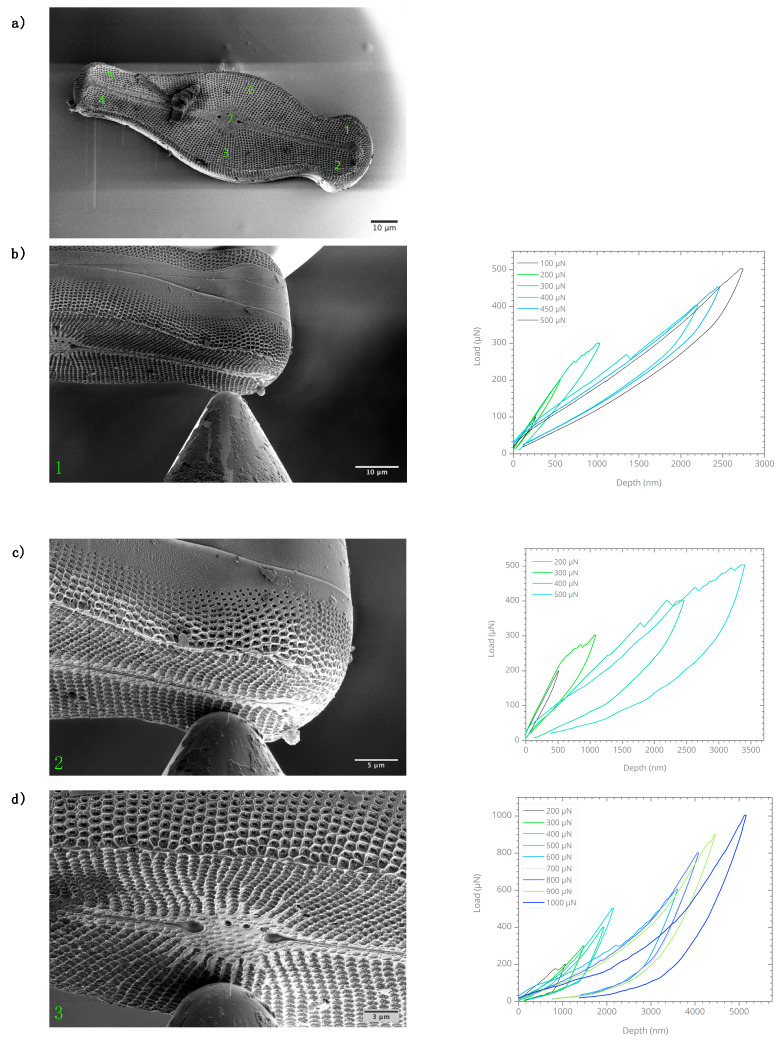
In situ micro-indentation experiments were conducted at seven different locations on the epivalve of *Didymosphenia geminata* frustule sample. (**a**) overview of the micro-indentation spots, (**b**–**h**) The micro-indentation spots and the obtained load-displacement curves.

**Figure 7 nanomaterials-10-00959-f007:**
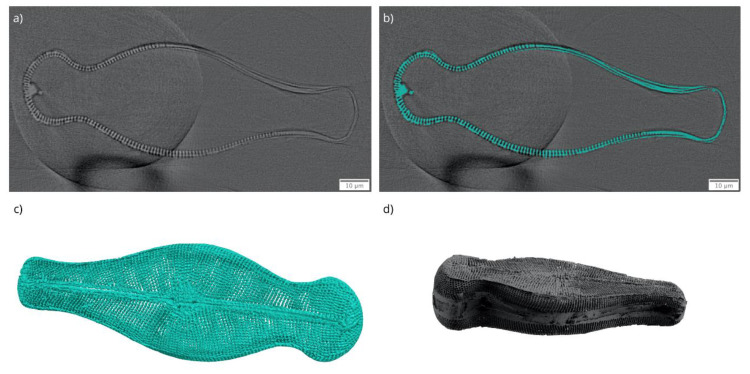
(**a**) Reconstructed nano-XCT data, (**b**) segmentation field, (**c**) consequent 3D visualization of segmented data, and (**d**) the surface mesh of studied *Didymosphenia geminata* frustule.

**Figure 8 nanomaterials-10-00959-f008:**
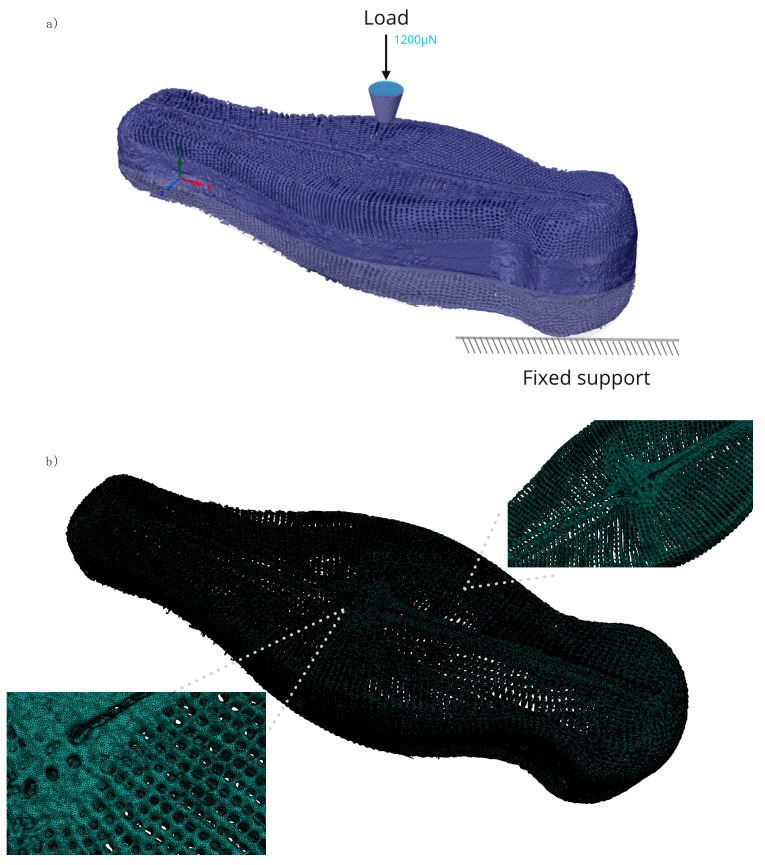
Finite element analysis model of the *Didymosphenia geminata* frustule: (**a**) boundary conditions used for replication of in situ micro-indentation experiments, (**b**) final FEM mesh used for the simulations that consists of 21.6 million tetrahedral elements and 1.2 million hexahedral elements with an average mesh quality of 0.95.

**Figure 9 nanomaterials-10-00959-f009:**
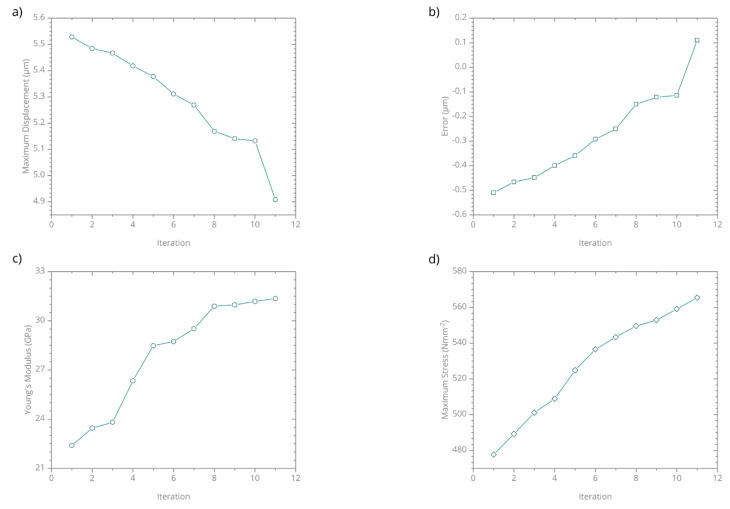
The iterative minimization of the deviation in the maximum displacement between the micro-indentation experiment and the simulation results: Iteration against (**a**) maximum displacement, (**b**) error, (**c**) Young’s modulus, (**d**) maximum stress at strain of 0.025.

**Figure 10 nanomaterials-10-00959-f010:**
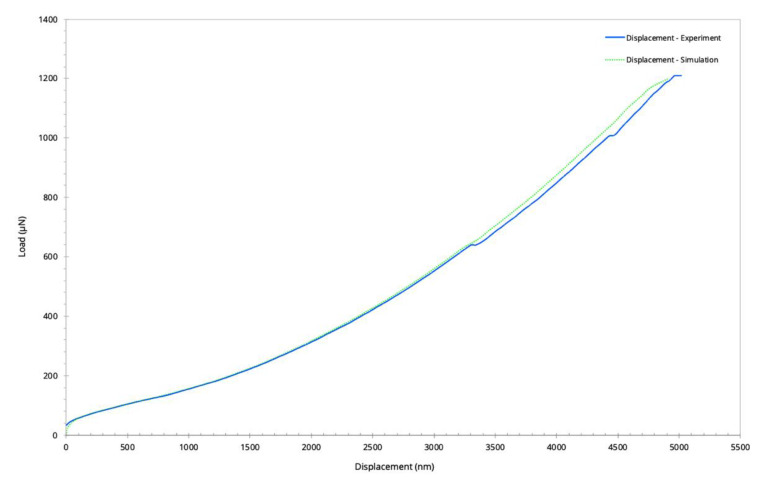
The obtained load-displacement curves from the micro-indentation experiment and the simulation during the loading.

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
