# Peer review of "Numerical and Experimental Study of the Mechanical Response of Diatom Frustules"

_nanomaterials, 2020, doi:10.3390/nano10050959_

Round 1

Reviewer 1 Report

The paper presents a FEM modeling of a micro-indentation test on a complex hierarchical biological porous structure, a diatom frustule. The complex 3D structure of the frustule has been obtained by X-Ray nanotomography. The main outcome of the study is the Young's modulus of the consituent material of the diatom, establish through a fitting of the maximum displacement measured experimentaly. Additional characterization by SEM and TEM are also presented.
The set of experimental results presented is challenging to obtain, rich, interesting but would have deserved a more in-depth analysis. In addition the FEM part (also challenging) needs clarifications and additional outputs for the reader to assess the quality of the simulation and obtain insight on the location of stress concentrations. For these reasons I recommend the publication in Nanomaterials only after major revision. More precisely, the authors should address the following points:

l19-21 : the sentence is misleading, the authors should remove "in the elastic-plastic region" and replace "simulations" by "elastic simulations" to conform to what I believe has been done.

l21-22 : Again, misleading: I am not very sure what the authors call maximum stress (see my comments further down) but stated like that in the abstract it will be interpreted as a yield stress or elastic limit stress and I believe this is not the case. For this reason, the authors should remove the mention of "maximum stress" and the corresponding value.

l23-25 : This last sentence of the abstract is also misleading about the content of the paper and should be at least tempered if not removed.

l61-62 : I'd like to point out a few additional references about silica plasticity at small scale, two MD simulations paper
E. C. Silva, L. Tong, S. Yip, K. J. Van Vliet, Size effects on the stiffness of silica nanowires, Small 2 (2006) 239–243
L. P. D´avila, V. J. Leppert, E. M. Bringa, The mechanical behavior and nanostructure of silica nanowires via simulations, Scripta Materialia 60 (2009) 843–846

and an experimental paper
R. Lacroix, G. Kermouche, J. Teisseire, and E. Barthel, “Plastic deformation and residual stresses in amorphous silica pillars under uniaxial loading,” Acta Materialia, 60 [15] 5555–5566 (2012).

l76-79 : some references would be useful here. I suggest C. Petit, S. Meille, and E. Maire, “Cellular solids studied by x-ray tomography and finite element modeling – a review,” Journal of Materials Research, 28 [17] 2191–2201 (2013).

l103-106 : The authors should keep this sentence for their conclusion.

l139-151 : The material model used should be specified, isotropic elastic or isotropic elasto-plastic (of which kind ?)

l220-221 : The author should be less affirmative on the origin of residual displacement. I believe the observed residual displacements might also have been caused by local damage/breakage of struts/ribs. Especially, the decrease in slope often observe on the curves of Fig6 might be a consequence of damage more than plasticity (the later should not modify the stiffness of the structure on reloading). The author state that "During the microindentation experiments, SEM images were recorded
continuously to track the deformation." SEM observations might provide interesting informations on this point, a selection of images during or after loading should be included and analysed in the paper.

l222 : non-linearities are also a consequence of the geometry of the test and of large displacements and deformations.

Fig. 6 : The text of the graphs (legend, axes,...) is not readable. The authors should increase the font size or rearrange the figure for the graphs to be bigger.

l231-232 : I agree there are elastic models for indentation of a bulk homogeneous material but in the case of a 3D complex structure at the same scale than the indenter there is no analytical solution possible.

l234-237 : What I understand from this part is that an elasto-plastic model was used ? But a few line after the author mention a purely elastic model ! The author should remove or rewritte this part.

l262-264 : For the sake of clarity, the authors might add "i.e. very limited residual displacements testity for neglegeable plasticty and damage, and allows the use of a purely elastic model"

Fig 8 : A close view of the mesh (at the scale of 2-3 ribs) should be given for the reader to judge how well the mesh reproduce the actual geometric details.

l269 : the Poisson's ratio used should be given.

l282 : the concept of "maximum stress at 0.025 equivalent von Misses strain" is not clear:
- which stress ? Von Mises ?
- why 0.025 strain and not something else ? where is the strain taken ?
- what is the purpose of this value ?

Fig 9: The evolution of values with iterations is not very useful/interesting in my opinion, reporting the final values is enough.

- The load displacement curve from the FEM analysis should be provided for the reader and compared to the experimental one to assess the quality of the fit.

- The authors should also provide the stress field (Von Mises) to show the various levels of stress localization. I think it will be advantageous to show close up view where the stresses are maximal.

- The deformed shape at maximum load should also be provided and might be compared to SEM images at maximum displacement (the authors claimed that they recorded SEM images during microindentation)

Author Response

Dear reviewer,

We thank you for your comments and suggestions to improve the quality of our manuscript.

Please see the attachment for our responses.

Thanks a lot.

Emre

Reviewer 2 Report

The paper "Numerical and experimental study of the mechanical 2 response of diatom frustules" is very interesting. The authors study the diatoms porous materials and measure mechanical properties, and they simulate them. That is very interesting because this approach can be made for another kind of porous material and it will be very interesting for readers from other areas like catalysis and adsorption. The paper is well written and discussed. I only suggest that the figure 6 must be divided or increase the size. In this actual size, it is impossible to see the graphics.

The experiments are very well done, the chosen points to do the measurements are perfect. The work is very interesting. Congratulations to the authors.

Author Response

Dear reviewer,

We thank you for your comments and suggestions to improve the quality of our manuscript.

Please see the attachment for our responses.

Thanks a lot.

Reviewer 3 Report

The manuscript entitled „Numerical and experimental study of the mechanical response of diatom frustules” authored by E. Topal, H. Rajendran, I. Zglobicka, J. Gluch, Z. Liao, A. Clausner, K. J. Kurzydlowski and E. Zschech shows combined approach to describe mechanical properties of Didymosphenia geminata frustule. I like the paper and I believe it is worth publishing after minor corrections.

Major issues:

  • Please clearly state novelty of the work. You cite some related works [25-31] but do not explain what is yours unique contribution.
  • The second biggest problem is that the paper does not have clearly stated “main story”. What is the most important thing that you try to convey and “who cares”? This should be clearly stated in the text. I assume that the most important thing is the new approach to extract the mechanical properties of biomaterials based on combined theoretical and experimental effort. If this is correct please underline how it is different from other works (not only on diatom frustules) and novel. However, if you think that the main finding is related to numbers you found (i.e. 31.8 GPa etc.) please clearly underline it and compare it to other cellular/porous structures (not only on diatom frustules).

Minor issues:

  • What are mechanical parameters expected for the amorphous bio silica itself [P2L69]?
  • P8L240 “due to its structural complexity it is not possible to design a representative complete Didymosphenia geminata frustule geometry in a CAD framework even if the exact feature sizes are known” – why? I don’t think it is impossible. Very hard and time consuming yes, but not impossible.
  • Part on TEM and EDX is poorly written (P6L181 to P6L196).
  • Quality of figures needs to be improved. There are several issues related to this. You have such beautiful 3D model of your frustule, please use it to improve your graphics. For instance, it would be nice to show the cross-section in Figure 2 and Figure 4a using 3D model. Also, 3D model in Figure 1 (upper left) and Figure 8b are poorly visible. I know that there is a lot of mesh vertices but you can surely represent it in better way. Also scale bars, especially in Fig 4 (a and b) are useless, please improve it. Figure 9 is strangely designed with a lot of space in between. It looks odd.

Author Response

(The authors gave the same response as above.)

Round 2

Reviewer 1 Report

The paper as been significantly improved and can be published in its present form. I'd just like to point out a typo, l327 Von Mises (and not Misses).